# Water Absorption Behavior of Oil Palm Empty Fruit Bunch (OPEFB) and Oil Palm Kernel Shell (OPKS) as Fillers in Acrylic Thermoplastic Composites

**DOI:** 10.3390/ma15145015

**Published:** 2022-07-19

**Authors:** Cristina E. Almeida-Naranjo, Vladimir Valle, Alex Aguilar, Francisco Cadena, Jeronimo Kreiker, Belén Raggiotti

**Affiliations:** 1Departamento de Ciencias de Alimentos y Biotecnología, Facultad de Ingeniería Química y Agroindustrial, Escuela Politécnica Nacional, Ladrón de Guevara E11-253, Quito 170517, Ecuador; cristina.almeidan@epn.edu.ec (C.E.A.-N.); alex.aguilar@epn.edu.ec (A.A.); francisco.cadena@epn.edu.ec (F.C.); 2Facultad de Odontología, Universidad de las Américas, Redondel del Ciclista Antigua Vía a Nayón, Quito 170124, Ecuador; 3Centro Experimental de la Vivienda Económica (CEVE)-CONICET, AVE, Igualdad 3585, Córdoba X5003BHG, Argentina; jerokreiker@gmail.com; 4Centro de Investigación, Desarrollo y Transferencia de Materiales y Calidad (CINTEMAC), UTN-FRC, Maestro M. López y Cruz Roja Argentina, Córdoba X5003BHG, Argentina; belenraggiotti@gmail.com

**Keywords:** oil palm waste, empty fruit bunch, kernel shell, acrylic resin, water diffusion

## Abstract

In recent years, the use of oil palm wastes has been an interesting approach for the development of sustainable polymer matrix composites. Nevertheless, the water absorption behavior of these materials is one of the most critical factors for their performance over time. In this study, the water uptake characteristics of acrylic thermoplastic matrix composites reinforced separately with oil palm empty fruit bunch (OPEFB) and oil palm kernel shell (OPKS) were evaluated through immersion test in distilled water. The specimens of both composites were manufactured using the compression molding technique at three temperatures (80, 100, and 120 °C) using different particle sizes (425–600 and 600–850 µm). The composites, before and after the absorption test, were characterized by means of Fourier transform infrared spectroscopy, thermogravimetry, and scanning electron microscopy. The evaluation was complemented by the application of the Fickian diffusion model. Overall results showed that water absorption capacity decreased at a higher processing temperature and a larger particle size. In particular, it was observed that the type of reinforcement also influenced both water absorption and diffusivity. OPKS/acrylic and OPEFB/acrylic composites reached a maximum absorption of 77 and 86%, with diffusivities of 7.3 × 10^−9^ and 15.2 × 10^−9^ m^2^/min, respectively. Experimental evidence suggested that the absorption mechanism of the composites followed a non-Fickian model (*n* < 1.0).

## 1. Introduction

In recent decades, ecological and environmental consciousness has triggered the use of renewable materials as reinforcements in composite development [1]. In polymer matrix composite production, exploiting natural fibers has raised great interest and become of importance due to their well-known characteristics, i.e., high availability, low cost, renewability, lightweight, compostability, and non-toxic [2,3,4,5]. Reinforcement of polymer matrices with natural fibers seems to be significant in lightweight and energy conservation applications. For instance, the use of those composites in the transport industry reduces fuel consumption and therefore the generation of greenhouse gases [6,7]. Specifically, Kline & Company forecasted an annual increase in the demand between 15 and 20% for natural fibers/woods as plastic reinforcement for automotive applications [5].

The use of all sorts of natural fibers as alternatives to replace synthetic fibers is quite economical, since the cost of composite elaboration can be reduced. However, the use of natural fibers from agroindustrial wastes, abundantly available and currently underutilized, is not only economically attractive but also environmentally responsible in terms of recycling and circular economy. Nowadays, some of the wastes being used as reinforcements are bagasse fiber, wheat husks, coir pith, rice husks, hazelnut shells, and oil palm fiber [1,8]. Oil palm (*Elaeis guineensis Jacq.*) is the most widely produced edible oil crop in the world, and it can be found in 42 countries with approximately 11 million ha [1]; nevertheless, its plantation is generating abundant lignocellulosic wastes such as empty fruit bunch and kernel shell [9]. Ecuador, as the world’s ninth largest exporter of oil palm, generates around 7 million tons of waste/year [10]. It is considered that less than 40% of these wastes are reused as soil conditioner/fertilizer or fuel for boilers, the latter of which has produced greenhouse gas emissions [8].

In particular, oil palm empty fruit bunch (OPEFB) and oil palm kernel shell (OPKS) are composed of lignin (14–31%), cellulose (24–65%), and hemicellulose (21–34%) [11]. These lignocellulosic components make oil palm wastes potential reinforcements for a number of different thermoplastic [12] and thermosetting [7] polymer matrices. For instance, it has been reported that the addition of modified OPKS powder to polyester increased the thermal degradation temperature (from 370 to 418 °C) compared to pure polyester or pure powder [13]. Moreover, nanosilica-hybridized OPKS in thermoplastic polypropylene composites improved tensile strength, impact strength, and elongation at break [14]. On the other hand, the use of OPEFB in thermosetting matrices increased mechanical integrity and acoustic absorption [15]. While in hybrid composites OPEFB improved the impact resistance, as well as the tensile and flexural strength [16]. In the case of composites hybridized with thermoplastic matrices, the flexural/tensile modulus and tensile/tear strength were improved [13].

Although thermosetting matrices in natural fiber composites have shown several functional applications, thermoplastic matrices offer other advantages, namely low processing cost, design flexibility, ease of molding complex parts, and recyclability. Likewise, several thermoplastic composites reinforced with natural fibers are flexible, tough, and have good specific mechanical properties [17], i.e., polypropylene composites reinforced with hemp fibers [18]. Over the last few years, systematic studies have narrowly dealt with acrylic matrix composites due to the versatility of acrylic polymers [19,20]. However, acrylic thermoplastic composites, as well as other polymeric matrices reinforced with natural fibers, have drawbacks such as limited processing temperature, lack of compatibility between fiber and matrix, low fire resistance, and high water absorption [7,17]. The latter is a critical parameter in the performance of these composites, since water absorption usually produces poor adherence between the matrix and fibers, poor stress transfer, matrix degradation, and dimensional instability [21,22,23]. The water absorption in thermoplastic matrix composites reinforced with natural fibers is mainly related to the chemical composition and microstructure of natural fibers [21]. The mechanism governing the movement of water molecules inside the composites is diffusion [21,24]. Fick’s law, due to its simplicity, is one of the models used to predict the diffusion mechanisms in composites. Nevertheless, the non-Fickian model or the intermediate (Fickian–non-Fickian) model could adjust to the water diffusion in composites [21,24,25]. According to the Fickian laws, the diffusion of water molecules in composites occurs in three stages. Initially, the water molecules flow into the microcracks of the polymer chains, followed by the movement of water molecules (capillary) into the cavities and voids that exist in the fiber–matrix interface, and finally, swelling of the composites occurs, which advances towards the microvoids of the matrix [21].

The literature focusing on this topic has established that hollow cavities, together with the irregular and porous surfaces of OPEFB and OPKS, affect the hygroscopic character of composites elaborated with these natural reinforcements [3,11,26]. OPEFB (±13 mm)/phenolic resin (50/50 wt.%) composites absorbed 22.29% of water, which was reduced by 6.89% when composites were mixed with cane bagasse (OPEFB = 15, bagasse = 35 wt.%) [23]. According to Ayu et al. (2020), poly(butylene) succinate/starch/glycerol composite reinforced with different amounts (8, 12, 16, and 20 wt.%) of OPEFB (300–600 µm) showed water absorptions between 3.7 and 5.5% [27]. On the other hand, polyester resin composites with different concentrations (10, 20, and 30 *v*/*v*.%) of OPKS (1.0–2.8 mm) showed absorption percentages between 0.7 and 1.1%, where the absorption increased directly with OPKS content [28]. A similar relationship was found using phenol formaldehyde and urea formaldehyde reinforced with oil palm trunks, composites that reached maximum percentages of water absorption of 36.2 and 46.0%, respectively, when using the resins at a concentration of 25%. It was determined that increasing the resin content from 25 to 75% decreased the water adsorption capacity by 20.0 and 25.2% for the urea formaldehyde and phenol formaldehyde resins, respectively. Nevertheless, Shehu et al. (2013) did not find a direct relationship between the content of polyester resin and OPKS (0–40 *w*/*w*. %), but they determined an increase in water absorption with decreasing particle size. In this case, the authors used particles of 75, 150, and 300 µm [29].

There are many differences in the water absorption patterns among the aforementioned studies, which could be associated with matrix microstructure, size/type/microstructure of reinforcement, processing conditions, and so forth. Although there has been a keen interest in studying composite materials with natural reinforcements, few investigations have focused on determining the water absorption capacity of thermoplastic polymer composites reinforced lignocellulosic fillers. It is thus of practical significance to comprehensively investigate the water absorption capacity of composites based on oil palm wastes and acrylic polymers. Specifically, the objective of this research was aimed at observing the influence of reinforcement type, filler size, and processing temperature on the water absorption behavior of acrylic thermoplastic composites reinforced with OPEFB and OPKS. In this case, the reinforcements and the composites were characterized using analytical and instrumental methods. Moreover, the water absorption mechanism was determined by fitting the obtained data to the Fickian model and calculating the transport coefficients.

## 2. Materials and Methods

### 2.1. Materials

The liquid acrylic thermoplastic resin (SINTACRIL A-292^®^) was purchased from Poliacrilart, Quito, Ecuador. The density and Brookfield viscosity (SP1, 12 rpm) of the resin at 25 °C were 1.06 ± 0.01 g/cm^3^ and 70 cP, respectively. Oil palm wastes (OPEFB and OPKS) were supplied by Teobroma, Alcopalma industry, located in the city of Quinindé, Ecuador (0°20′ N 79°29′ W). 

### 2.2. Preparation of Composites

Prior to elaborating composites, OPEFB and OPKS were dried at room temperature and separately ground in two cycles using the SHINI blade mill, model SG-2348E (Ningbo, China), for OPEFB and the THOMAS WILEY knife mill, model 3379-K05 (Swedesboro, NJ, USA), for OPKS. The oil palm wastes were then sieved in two size groups: 425–600 and 600–850 µm. Subsequently, they were dried at 103 °C for 3 h. In addition, proximal analysis of conditioned fillers was performed by means of standardized testing. Moisture (ASTM D 4442), extractives (ASTM D 1107/ASTM D 1110), lignin (ASTM D 1106), hemicellulose and cellulose (ASTM D 1109), and ash content (ASTM D 1102) were determined.

As a second step, acrylic resin and conditioned fillers were mechanically mixed at 400 rpm for 30 min at room temperature. Excess resin was squeezed out of the embedded fillers, which were dried at 103 °C for 3 h. Imbibed fillers were then processed using the LAB TECH compression molding machine, model LP-S-50 (Mueang Samut Prakan, Thailand). Three processing temperatures (80, 100, and 120 °C) were applied in the molding press, while the pressure was kept constant at 150 bar for 40 min. OPEFB/acrylic and OPKS/acrylic composites were separately obtained at different filler sizes and molding temperatures in the form of sheets measuring 150 mm × 150 mm × 2 mm [30].

### 2.3. Material Characterization

In order to determine the influence of reinforcement type, filler size, and processing temperature over the water absorption of the composites, infrared, thermal, and morphological characterization were performed before and after the absorption test. In doing so, functional groups were identified by Fourier transform infrared spectroscopy (FTIR) in attenuated total reflection mode using the JASCO spectrometer, model FT/IR-C800 (Tokyo, Japan). Twenty scans were performed in the range between 4000 and 600 cm^−1^, with a resolution of 4 cm^−1^. Furthermore, thermogravimetric analysis (TGA) was performed using the SHIMADZU thermobalance, model TGA-50 (Kyoto, Japan), in the range of 20 to 600 °C, with a heating rate and nitrogen flow of 10 °C/min and 50 mL/min, respectively. In order to study the morphological features of the composites, scanning electron microscopy (SEM) was performed by using the ASPEX scanning electron microscope, model PSEM eXpress (Billerica, MA, USA), with a working distance of 20.4 mm and an acceleration of 15 kV.

### 2.4. Absorption Assays

The water absorption test was adapted from the ASTM D5529 and UNE-EN-2378 standards. The sheets were cut into 50 mm × 50 mm specimens. The samples were dried for 8 h at 70 °C, cooled in a desiccator, and weighed. Five specimens of each formulation were immersed in a distilled water bath (pH = 4.84 ± 0.36; EC = 4.08 ± 0.24 µS/cm) at 23 °C. The volume of the water bath was controlled and measured at 1 L throughout the test. The specimens were removed from the water according to the time intervals suggested by the UNE-EN-2378 standard, until the percentage of the water absorption as a function of the square root of time was constant. Afterwards, the samples were dried, weighed, and immersed again in the distilled water bath. At the end of the test, the samples were dried (24 h at 70 °C) and weighed. The moisture content absorbed by each specimen was calculated from its weight before and after the absorption for each time interval, as follows:(1)$$M\left(\%\right)=\frac{w_{t}-w_{o}}{w_{o}}\times 100$$
where *M* is the moisture content (%), *w_t_* is the specimen weight at time *t* (g), and *w_0_* is the specimen weight before absorption (g).

Moreover, the dimensions of the specimens were measured before and after the water absorption test in order to determine the thickness swelling, according to Equation (2).
(2)$$\;\;\;\;\mathrm{Thickness}\;\mathrm{swelling}\;\left(\%\right)=\frac{TS_{t}-TS_{o}}{TS_{o}}\times 100$$
where *TS_t_* is the specimen thickness after the absorption test (mm), and *TS_o_* is the specimen thickness before the absorption test (mm).

The kinetics and diffusion mechanisms of the water in the composites were analyzed by fitting the obtained data to the non-linear Fickian diffusion model, according to Equation (3):(3)$$\frac{M_{t}}{M_{m}}=k\times t^{n}$$
where *M_t_* is the moisture absorption at time *t*, *M_m_* is the moisture absorption at saturated condition, *k* is a characteristic constant of the composites, and *n* is an expansion exponent that describes the mode of the penetrant transport mechanism [31]. 

The diffusion coefficient (*D*) is an important parameter in the Fickian model, which indicates the ease with which the water molecules could penetrate the composites. *D* was calculated with Equation (4) [32]:(4)$$D=\pi {\left[\frac{h}{4\times M_{m}}\right]}^{2}{\left[\frac{M_{2}-M_{1}}{\sqrt{t_{2}}-\sqrt{t_{1}}}\right]}^{2}$$
where *h* is the average thickness of the composites, *t*_1_ and *t*_2_ are the times in the linear portion of the curve, and *M*_1_ and *M*_2_ are the percentages of moisture absorption at *t*_1_ and *t*_2_, respectively.

Another relevant parameter in the kinetics of water absorption is the absorption coefficient (*S*), which is related to the saturation point of water absorption. *S* was determined with Equation (5) [33]:(5)$$S=\frac{W_{m}}{W_{p}}$$
where *W_m_* is the mass of absorbed water at equilibrium swelling, and *W_p_* is the mass of the composite.

The permeability coefficient (*P*), which indicates the net effect of both the diffusion and the absorption coefficient, was obtained using Equation (6) [33]:(6)P=D×S

For a better understanding of the influence of experimental variables on water absorption, one-way analysis of variance (ANOVA) was performed. Parametric analysis of variance was used for normal data sets (Tukey’s test) and nonparametric evaluation (Kruskal–Wallis) for non-normal distributions. The statistical analysis of the non-linear Fickian model considered parameters of central tendency and spread. In addition, the correlation coefficient (*R*^2^) and the chi-square (*χ*^2^) were calculated (Equations (7) and (8)) to assess the fit of the water absorption data to the Fickian model.
(7)$$R^{2}=1-\frac{\sum {\left(M_{m,\;exp}-M_{m,\;cal}\right)}^{2}}{\sum {\left(M_{m,\;exp}-M_{m,\;mean}\right)}^{2}}$$
(8)$$\chi^{2}=\sum \frac{{\left(M_{m,\;exp}-M_{m,\;cal}\right)}^{2}}{M_{m,\;cal}}$$
where *M_m,exp_* is the moisture absorption under saturated conditions, *M_m,cal_* is the moisture absorption calculated using the solver tool (mg/g), and *M_m,mean_* is the mean of the *M_m,exp_* values.

## 3. Results and Discussion 

### 3.1. Material Characterization 

#### 3.1.1. Reinforcement Characterization

The physical–chemical characterization and proximal analysis of oil palm wastes are presented in Table 1. The obtained results were in accordance with the ranges presented by different authors [3,11,23,34]. Furthermore, lignocellulosic content (lignin + cellulose + hemicellulose) in OPEFB and OPKS was 97.07% and 92.07%, respectively. These results were similar to other natural fillers such as flax, jute, and abaca, of which the reported values were 92%, 90%, and 97%, respectively. It should be noted that the high content of polysaccharides is characteristic of agroindustrial wastes [3]; from the perspective of water absorption, some polysaccharides have an affinity for water molecules due to the presence of hydrophilic groups in their structure [35].

The polysaccharide involved in water absorption is hemicellulose (the higher the hemicellulose content, the greater the water absorption) due to its hydrophilic character. On the other hand, higher lignin content makes lignocellulosic material less hygroscopic. In this case, the hemicellulose content in OPEFB and OPKS differed by 1.80%, and the lignin content differed by 29.82%, suggesting that OPEFB will have a higher water absorption capacity [36,37].

SEM images of OPEFB and OPKS are presented in Figure 1a,f. On the whole, porous and heterogeneous surfaces were clearly observed in both fillers [26,38]. Moreover, the presence of microfibers was observed in OPEFB, which was in good agreement with the results reported by Phreecha et al. (2019) [35], who noticed that these morphological characteristics made both fillers good sorbent materials [38].

#### 3.1.2. Composite Characterization

The SEM images of the composites that presented the highest water absorption capacity before and after the absorption test are presented in Figure 1. In the micrographs, prior to the absorption process, voids were observed, and they were more noticeable in the OPEFB composites than in the OPKS composites because of the lower dispersion of the reinforcement in the acrylic matrix [23]. Voids were produced by insufficient adherence/wetting between the polymer matrix and the reinforcements due to the hydrophobic and hydrophilic character they present [14,15]. However, Bin Bakri et al. [15] associated the presence of voids with the manufacturing process of the composites, since the process of mixing the acrylic resin with the oil palm wastes could enter the air, which could be trapped. It is observed that, after the absorption process, there were morphological changes in the composites, which were associated with water molecules entering their structure. In the OPEFB/acrylic composites, there was fiber swelling, while in both composites, the voids increased in size and quantity. This suggested that OPEFB/acrylic composites had greater absorption capacity not only due to their greater porosity and microfiber presence (Figure 1a) but also due to the seemingly lower adhesion between the fibers and the acrylic matrix [35].

The effects of water absorption on the infrared characteristics of the studied composites are shown in Figure 2. It should be noted that the water–composite interactions were mainly associated with the hydrogen bonds between the water molecules and the carboxyl/hydroxyl groups of the lignocellulosic components from the OPEFB and OPKS fillers. The FTIR spectra of the composites after the absorption test (colored lines) showed very pronounced changes between 3700 and 3000 cm^−1^ with respect to the composites before the absorption test (black dashed lines). This behavior was related to the stretching vibration of the O-H groups of cellulose and hemicellulose. The stretching of the C-H and CH_2_ groups of cellulose and hemicellulose was modified in the bands around 2955 and 2845 cm^−1^, respectively. Another significant impact of the interaction between water and composites was observed in the band at 1640 cm^−1^, which was characteristic of the O-H group bending vibration of the adsorbed water. The intensity of the bands between 1550 and 1509 cm^−1^ increased because of the interaction with the aromatic ring of lignin. The changes (less pronounced) observed in the bands near 1430 cm^−1^ were attributed to the carboxylic acid of pectin and COO− vibration, while the changes around 1380 cm^−1^ were produced by the C-H bending of cellulose and hemicellulose. In addition, there were differences in bands at 1220, 1140, and 1024–980 cm^−1^; these changes were observed due to the presence of C-O of acetyl in pectin or hemicellulose, the C-O-C skeletal vibration of cellulose, and the coupling interaction between the COOH stretching vibrations of cellulose, respectively [35,39].

The thermal behavior of the composites before and after the absorption test is presented in Figure 3 and Figure 4. The thermograms showed four stages of weight loss that occurred around the following temperature ranges: 70–150 °C, 250–350 °C, 350–410 °C, and 410–600 °C, with temperatures around 280 and 400 °C being the temperatures at which the highest rate of mass loss occurred for the composites. The first stage presented the lowest weight losses (0.66–3.22% and 2.47–4.13% for the OPEFB and OPKS composites, respectively), which was attributed to water evaporation (intra-/intermolecular dehydration reactions) [9,26]. The second stage corresponded to hemicellulose degradation, obtaining losses between 14.52–26.43% (before the test) and 9.91–13.29% (after the test) for OPEFB and 10.29–23.89% (before the test) and 10.19–22.79% (after the test) for OPKS [9,31]. The difference in weight losses, before and after the water absorption test, in the OPEFB/acrylic composites was produced because the soluble fraction of hemicellulose could be solubilized in water during the absorption process. It is, however, worthy of note that the behavior of the OPKS/acrylic composites was different due to the fact that the fraction soluble/insoluble of hemicellulose in water depends on the type of reinforcement/species (biomass source) [40].

The third transition stage, with losses between 39.88 and 62.04% for the OPEFB/acrylic and 46.27 and 63.13% for the OPKS/acrylic composites, was the main phase of thermal degradation for the composites, which was associated with the decomposition (carbonization and byproduct volatilization) of cellulose [9]. In this case, the differences in weight losses were generated by the different crystallinity indices of cellulose [9,24]. The final stage was characterized by slow weight losses associated with lignin degradation [9,31]. The OPEFB/acrylic composites after the absorption process had considerable losses (24.37–37.89%), while the OPEFB/acrylic composites before the absorption process and the OPKS/acrylic composites before and after the absorption process showed lower losses (5.60–14.54%). The high weight losses in the OPEFB/acrylic composites after the water absorption process could be a consequence of weak bonds between the lignin of the natural reinforcement and the polymer matrix, so the absorption of water weakened these bonds even more, thus producing greater degradation [41].

### 3.2. Absorption Assays

Figure 5 shows the water absorption of composites as a function of time. The amount of absorbed water by the composites increased with the immersion time, behavior that is common in several composites reinforced with natural fibers [42,43]. One of the mechanisms of water transport inside composites is the diffusion of molecules within the microgaps/voids between acrylic polymer chains and OPEFB/OPKS [31,44]. The water molecules diffuse into the composites and adhere to the hydrophilic groups of the natural reinforcement, forming intermolecular hydrogen bonds (hydrogen bridges) with the functional groups of the fillers (hydroxyl and other oxygen-containing groups) [23,31]. OPEFB/acrylic and OPKS/acrylic composites of size 425–600 µm showed higher water absorption, 85.92 ± 0.68% at 1680 h and 76.92 ± 0.68% at 1704 h, respectively. Nonetheless, the absorption capacity of composites with the aforementioned filler size did not display significant differences regarding the processing temperature. However, in the OPEFB/OPKS composites with the largest fill size, the processing temperature significantly affected their water absorption capacity. The differences in the water uptake between the OPEFB and OPKS composites of size 425–600 µm were due to the fact that OPEFB fibers have nonuniform lengths compared to OPKS particles. Thus, the obtained results suggested that the nonuniform OPEFB lengths produced lower fiber–matrix interface compatibility, formation of voids and cavities, and high water absorption capability because the latter is a function of the size and shape of the reinforcement [33]. Moreover, the higher water absorption of the OPEFB composites was produced by fiber microstructure, which is markedly different from that of the OPKS (Figure 1), i.e., the OPEFB is formed by microfibers and lacunas. The lacunas are located in the central part of the fiber and facilitate the transport of water in the fiber radial direction [45]. In addition, it can be seen that the OPKS (600–800 µm) composites did not reach equilibrium at 1704 h; however, it was tested up to this time to establish a comparison with the OPEFB and OPKS (425–600 µm) composites that reached equilibrium at 1008 h; that is, there were no significant differences in water absorption at times greater than 1008 h in any of the OPEFB and OPKS (425–600 µm) composites.

On the other hand, all OPEFB/acrylic composites (600–850 µm) at 24 h of exposure reached between 22.61 and 24.40% of water absorption. Ramlee et al. [23] pointed out the water absorption after 24 h of exposure at 22.29% of the phenolic matrix composite reinforced with OPEFB (13 mm, 50/50 wt.%). However, the final absorption capacity reported by Ramlee et al. [23] differed (up to 7.27%) from the OPEFB/acrylic composites (425–600 µm) and even more so from the OPKS/acrylic composites in which, at 24 h, the water absorption was between 7.73 and 15.58%. Therefore, it was checked again that the size and type of the reinforcement influenced the absorption behavior of the composites.

The particle size significantly influenced the water absorption capacity of the composites. The samples with a filler size of 425–600 µm presented around 80% of water absorption and those with particles/fibers of 600–850 µm around 60%. The smaller particles exhibited higher water uptake due to their larger surface area (more available active sites) and interparticle friction, leading to smaller particle flow velocities [43]. The surface morphology of OPEFB and OPKS also influenced the absorption behavior; that is, greater porosity favored water uptake [43]. Furthermore, the increase in porosity led to boost the permeability and reduced the resistance to water flow in the porous media of the fillers [46,47].

Other factors such as the chemical composition of the reinforcements, the structure of the interfacial junction of the fillers, and the manufacturing parameters (processing conditions) could affect the water absorption of composites [23]. In this case, the processing temperature was another parameter that influenced the water absorption capacity in both composites. The water absorption decreased with the increase in the processing temperature, but there were no significant differences (*p* > 0.05) except between the OPKS/acrylic compounds, with the smaller particle size manufactured at 80 and 120 °C. The increase in processing temperature produced a macromolecular rearrangement and loss of water of the fillers with the acrylic matrix both in the cross-section and on the surface (higher degree of compaction/better fiber–acrylic mechanical bond) [37,44]

The water absorption capacity of composites reinforced with OPEFB or OPKS in previous studies was lower than that obtained in this investigation. For instance, the OPKS (30 wt.%)/orthophthalic unsaturated polyester and OPKS (10–40 wt.%)/polyester composites showed absorption capacities around 1.0 and 2.5%, respectively [28,29], and the OPEFB (8–20 wt.%)/polybutylene succinate composite between 4.2 and 5.5% [27]. The higher absorption capacity of the OPEFB/OPKS composites was due to the fact that the acrylic matrix used in this work, alone adsorbed about 31% of water, which was much higher than that of the matrices used in the investigations mentioned above, since they reached absorptions between <1.0 and 3.7% [27,28,29].

Table 2 details the parameters of the Fickian model and the kinetic/transport coefficients of the composites. The expansion exponent (*n*) presented values less than 1.0, which indicated that non-Fickian diffusion occurred. In addition, the *R*^2^ value was close to 1, and the values of *χ*^2^ were relatively low (between 0.18 × 10^−3^ and 7.41 × 10^−3^). This behavior has been observed in other polymer composites (i.e., epoxy composites). One characteristic of composites with non-Fickian behavior is that they take a long time to reach equilibrium in the water absorption test, which was reported in this study [48]. In the non-Fickian behavior, the water movement inside the composites was produced by the combination of both the water diffusion through the fillers and the movement of the acrylic polymer chains. The OPKS/acrylic composites had higher *n* values (around 0.5) than the OPEFB/acrylic composites, which suggested that the OPKS had more available sites for further absorption, so the diffusion process was faster, and their behavior was closer to the Fickian model [21,31].

The coefficient *D* values indicated that the increase in the processing temperature enhanced the resistance to water absorption (lower *D* value). The coefficient *D* increased exponentially with respect to the water uptake due to the lower binding energy among the active sites and the bound water molecules when there was high moisture content [22]. Nonetheless, a higher coefficient *D* could also indicate a higher void content in the composite; thus, water had more pathways to diffuse into the composite. Therefore, as stated before, the increase in the processing temperature improved the adhesion between the acrylic matrix and the fillers, a trend that slowed down the diffusion processes, since there were fewer spaces in the interfacial region [32,33]. The absorption (*S*) and permeability (*P*) coefficients presented the same behavior as the coefficient *D*; that is, they decreased when the processing temperature increased. This relationship between transport coefficients was in agreement with other research [33].

The swelling results of the composites during the water absorption process are presented in Table 3. The thickness swelling of the OPEFB/acrylic composites (between 15.93–29.08%) was less than that of the OPKS/acrylic composites (between 31.87–81.39%), which verified the influence on the water absorption capacity of the composites by type, morphology, size, and microstructure of the reinforcement, as well as by the formation of voids. Moreover, the thickness swelling of composites increased with decreasing processing temperature because the water absorption capacity of composites is directly related to the swelling phenomenon. This process produced swelling in the three dimensions of the composites; however, it should be noted that the thickness increased considerably in relation to the width and length. Additionally, water uptake in natural fibers was influenced by fiber swelling and changes in the fiber density (after water absorption) [44].

## 4. Conclusions

The water absorption behavior of acrylic thermoplastic composites reinforced with different filler sizes from oil palm wastes (OPEFB and OPKS) and manufactured at different processing temperatures was determined. The absorption capacity was related to the characteristics of the reinforcement, the presence of functional groups, and the morphology of the composites. The characterization of the reinforcements and the composites showed the presence of polysaccharides, functional groups (mainly carboxyl and hydroxyl), porosity, and voids. The water absorption capacity of the composites was mainly influenced by the morphology/size of the reinforcements, as well as by the manufacturing temperature. In particular, the water absorption increased as particle size and processing temperature decreased. The OPEFB/acrylic composites with the smallest particles (425–600 µm) presented the highest absorption capacity, 85.92 ± 0.68% at 1680 h. Moreover, the water absorption in the OPEFB/acrylic and OPKS/acrylic composites was dominated by the non-Fickian mechanism (*n* < 1). The transport coefficients were higher (between 1.5 and 4.5 times) for the OPEFB/acrylic composites, indicating that they were the ones with the highest absorption capacity. The results reported in this work provided an insight into the water absorption mechanisms of the OPEFB/acrylic and OPKS/acrylic composites. In future research, an attempt should be made to evaluate another type of acrylic resin (with less absorption capacity), and different concentrations of OPEFB and OPKS. In addition, an analysis of the mechanical integrity after the water absorption test should be carried out, since it could be of help to further specify the application areas of these composites.

## Figures and Tables

**Figure 1 materials-15-05015-f001:**
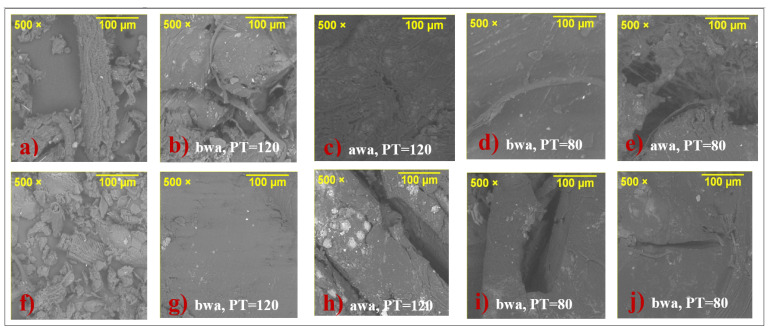
SEM images of (**a**) OPEFB, (**f**) OPKS, (**b**–**e**) OPEFB/acrylic composite, and (**g**–**j**) OPKS/acrylic composite. PT = processing temperature (°C), bwa = before water absorption test, and awa = after water absorption test.

**Figure 2 materials-15-05015-f002:**
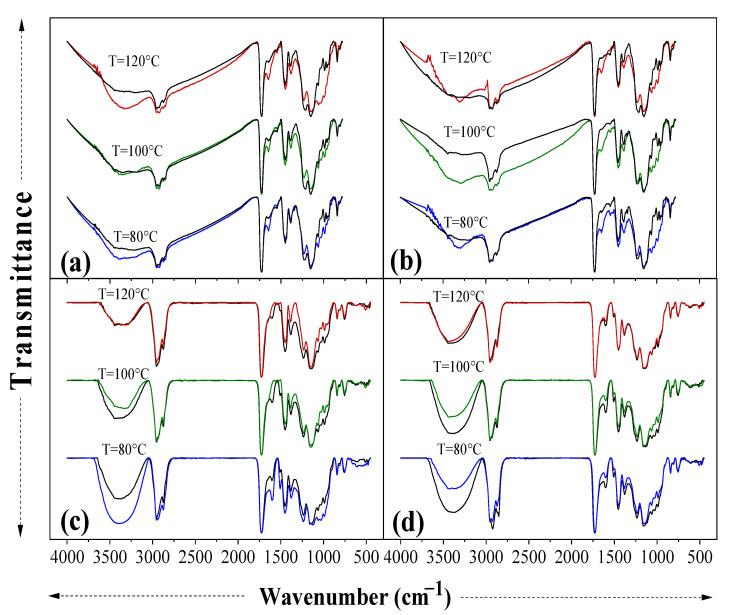
FTIR spectra before and after the water absorption test. OPEFB/acrylic composite: (**a**,**b**), and OPKS/acrylic composite: (**c**,**d**). (**a**,**c**) Size range=425–600 µm and (**a**,**b**) size range= 600–850 µm. Black dashed lines = control specimens (before the absorption test); colored solid lines = specimens after the water absorption test.

**Figure 3 materials-15-05015-f003:**
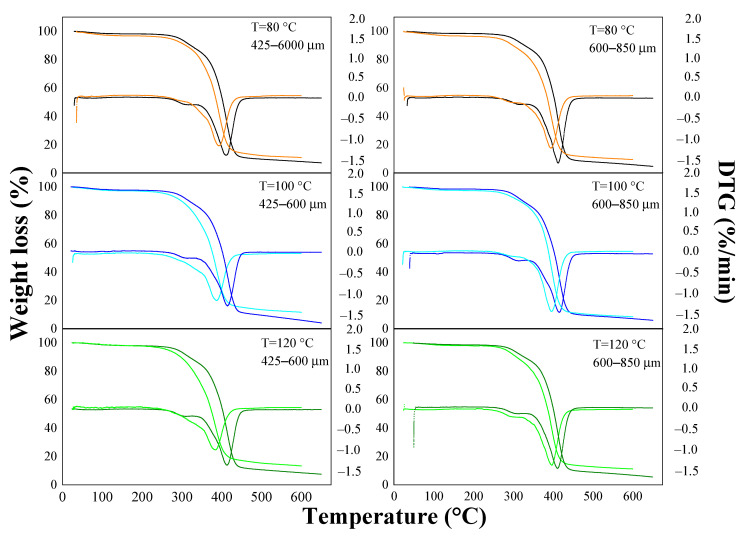
Thermal behavior of OPEFB/acrylic composites. Orange, cyan, and green light lines = before test; black, blue, and green lines = after test.

**Figure 4 materials-15-05015-f004:**
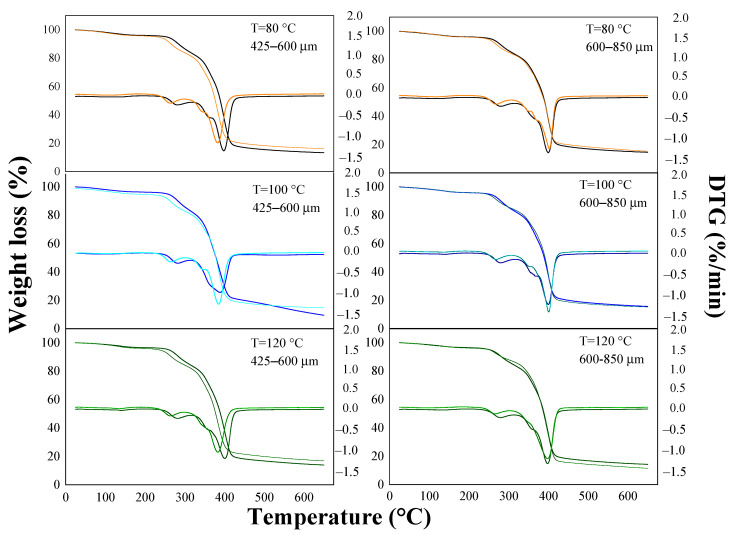
Thermal behavior of OPKS/acrylic composites. Orange, cyan, and green light lines = before test; black, blue, and green lines = after test.

**Figure 5 materials-15-05015-f005:**
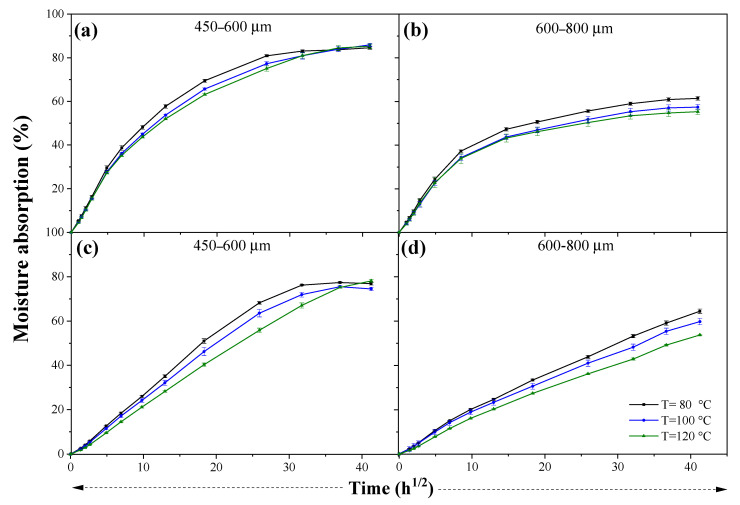
Experimental moisture absorption curves of OPEFB/acrylic composite (**a**,**b**) and OPKS/acrylic composite (**c**,**d**).

**Table 1 materials-15-05015-t001:** Physical and chemical characteristics of oil palm wastes.

Parameter	OPEFB	OPKS
Shape	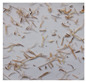	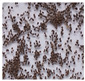
Fibers	Particles
Lignin (%)	19.12 ± 0.42	48.94 ± 0.51
Cellulose (%)	49.63 ± 0.64	24.96 ± 0.60
Hemicellulose (%)	21.32 ± 0.50	23.12 ± 0.20
Extractives (%)	1.46 ± 0.09	2.97 ± 0.11
* Moisture (%)	7.60 ± 0.47	8.80 ± 0.75
Ashes (%)	6.23 ± 0.19	8.69 ± 0.01

* It is the only parameter determined on a wet basis.

**Table 2 materials-15-05015-t002:** Parameters of the Fickian model and kinetic and transport coefficients of composites.

Filler	Particle Size (µm)	Processing Temperature (°C)	Constants for Fick Model				Transport Coefficient		
			Swelling Exponent (*n*)	Characteristic Constant (*k*)	*R* ^2^	*χ*^2^ × 10^−3^	*D* × 10^−9^ (m^2^/min)	*S* (g/g)	*P* × 10^−9^ (m^2^/min)
OPEFB	425–600	80	0.270	0.151	0.963	6.00	15.332	0.454	6.959
		100	0.285	0.136	0.976	4.06	11.692	0.447	5.231
		120	0.290	0.127	0.980	3.11	10.175	0.448	4.555
	600–850	80	0.249	0.175	0.961	6.64	8.583	0.371	3.182
		100	0.254	0.169	0.961	6.66	7.638	0.356	2.719
		120	0.246	0.179	0.957	7.41	7.532	0.348	2.622
OPKS	425–600	80	0.394	0.060	0.975	4.52	7.196	0.433	3.113
		100	0.410	0.053	0.981	3.34	6.852	0.418	2.868
		120	0.465	0.034	0.995	0.77	5.544	0.401	2.226
	600–850	80	0.424	0.047	0.998	0.24	6.106	0.347	2.120
		100	0.420	0.046	0.999	0.18	5.041	0.325	1.639
		120	0.440	0.040	0.998	0.27	5.138	0.300	1.541

**Table 3 materials-15-05015-t003:** Thickness swelling of composites.

Filler	Particle Size (µm)	Processing Temperature (°C)	Thickness Swelling (%)
OPEFB	425–600	80	15.93 ± 2.19
		100	22.66 ± 3.04
		120	17.14 ± 2.25
	600–850	80	29.08 ± 1.37
		100	22.10 ± 2.01
		120	19.73 ± 1.00
OPKS	425–600	80	54.08 ± 6.25
		100	37.68 ± 2.64
		120	31.87 ± 1.25
	600–850	80	81.39 ± 4.97
		100	58.19 ± 6.11
		120	39.29 ± 2.73

## Data Availability

Not applicable.

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
