# Peer review of "Water Absorption Behavior of Oil Palm Empty Fruit Bunch (OPEFB) and Oil Palm Kernel Shell (OPKS) as Fillers in Acrylic Thermoplastic Composites"

_materials, 2022, doi:10.3390/ma15145015_

Round 1

Reviewer 1 Report

In polymer matrix composite production, exploiting natural fibers has raised great interest and become of importance due to their well-known characteristics, i.e. high availability, low cost, renewability, light-weight, compostability, and non-toxic. In recent years, natural fiber reinforced polymer composites have gained much attention over synthetic fiber composites because of their many advantages such as low-cost, light in weight, non-toxic, non-abrasive, and biodegradable properties. On the contrary, natural fibres have some disadvantages, such as high moisture absorption due to repelling nature, low wettability, low thermal stability, and quality variation which lead to the degradation of composite properties. In this manuscript, the water uptake characteristics of thermoplastic acrylic matrix composites reinforced separately with oil palm empty fruit bunch and oil palm kernel shell were evaluated. The topic is important, the results are interesting and the methodology followed is appropriate, while the content falls well within the scope of this Journal. In general the paper makes fair impression and my recommendation is that it merits publication in this Journal, after the following major revision:

  1. The detailed literature review indicates efforts made by the authors. The coherence of the related work, however, is still not clear. It may help the authors by answering the following questions: Why are these works relevant? Which specific problems were addressed? How are the previous results related with the latest work? What are the outstanding, unresolved, research issues? Which of them has been solved by the proposed study? Answering the questions leads to the novelty of the proposed work naturally. I think this is essential to keep the interest of the reader.
  2. Materials and Methods part, Although the results look “making sense”, the current form reads like a simple lab report. The authors should dig deeper in the results by presenting some in-depth discussion.
  3. Natural fibers reinforced polymer composites were first introduced in 1908, by combining lignocellulosic fibers with phenolic resin. The natural fibers have been widely used in the industry. The natural fibers are found to serve in many practical applications, such as porous materials, (see [Powder Technology, 2019, 349:92-98; International Journal of Heat and Mass Transfer, 2019, 137:365-371). Authors should introduce some related knowledge to readers. I think this is essential to keep the interest of the reader.
  4. Overall results showed the influence of processing temperature, particle size, and reinforcement type over water absorption of composites. Particularly, it was observed that water absorbed and diffusivity of OPKS/acrylic composites were lower comparing with OPEFB/acrylic composites samples. Moreover, composites elaborated with OPKS and OPEFB showed water absorption up to 77 and 86%, respectively. This behavior was produced at higher processing temperature and larger particle size. Experimental evidence suggested that the absorption mechanism of the composites followed a nonFickian model. The authors should give some explanation on above conclusions and data.
  5. In Fig 4, the authors should give the explanations for the difference of data collected from different samples.
  6. Please expand the motivation, the problem context, clarify the problem description, and (if possible) add specific objectives.
  7. Please, expand the conclusions in relation to the specific goals and the future work.
  8. English grammar and syntax has to be checked carefully throughout the manuscript. There are several grammatical mistakes in the manuscript and it is very difficult to follow anything if they are not corrected.

Author Response

Dear Reviewer

We would like to thank you for the comments and suggestions for the manuscript entitled:

“Water absorption behavior of oil palm empty fruit bunch (OPEFB) and oil palm kernel shell (OPKS) as fillers in acrylic thermoplastic composites”. Following the suggestions received, we have modified the manuscript, added the suggested references, and carefully revised the complete document. We believe that our manuscript has improved substantially, and conveys in a better manner some key points of our research. 

Best regards,

Vladimir Valle

Reviewer 2 Report

Thank you very much for enabling the review of this manuscript.

The article presents a detailed analysis of kinetics of water absorption of  acrylic thermoplastic composites with a filler from natural waste arising on the oil palm plantations.

The tested materials should be included in polymer composites with filters of plant origin. In the literature you can find reports on the subject. However, the authors performed a reliable analysis of current knowledge and justified the purpose and reasons for the conducted research.

The overall impression after reading the presented work is good. I read the article with pleasure and curiosity. The obtained research results were correctly discussed and analyzed.

I have some comments:

- The description of the research results shows that a statistical analysis was performed for the results obtained (VV.:315, 338). There is no methodology of what statistical test was used and what value of significance level assumed, what number of samples was studied.

- I recommend writing variables with italics.

- I recommend changing the variable T (?ℎ??????? ????????) to a different value - T is reserved for temperature.

- Fig 1. - Please correct, complete the photo descriptions

- Please supplement the research methodology with the exact shape of the form used in the compression moulding.

Table 2 description - Does the description apply to waste or composites? – Please improve this description.

- I have quite a serious attention regarding the swelling curve for OPKS/ACRYLIC (600-800)- In my opinion the value of Moisture Absorption at Saturated Condition has not been achieved. This should be included in the text and marked in the results.

To sum up, I think that the article is suitable for publication after small editions.

Author Response

(The authors gave the same response as above.)

Round 2

Reviewer 1 Report

It is ok.

Author Response

We would like to thank you for the comments and suggestions for the manuscript entitled:

“Water absorption behavior of oil palm empty fruit bunch (OPEFB) and oil palm kernel shell (OPKS) as fillers in acrylic thermoplastic composites”. Following the suggestions received, we have modified the manuscript, added the suggested references, and carefully revised the complete document. We believe that our manuscript has improved substantially, and conveys in a better manner some key points of our research.
